# Species Composition of Parasitic Mites of the Subfamily Picobiinae (Acariformes: Syringophilidae) Associated with African Barbets (Piciformes: Lybiidae) [note 1]

**DOI:** 10.3390/ani13122007

**Published:** 2023-06-16

**Authors:** Bozena Sikora, Mathieu Mahamoud-Issa, Markus Unsoeld, Martin Hromada, Maciej Skoracki

**Affiliations:** 1Department of Animal Morphology, Faculty of Biology, Adam Mickiewicz University, Uniwersytetu Poznańskiego 6, 61-614 Poznań, Poland; bozena.sikora@amu.edu.pl; 2Department of Behavioural Ecology, Faculty of Biology, Adam Mickiewicz University, Uniwersytetu Poznańskiego 6, 61-614 Poznań, Poland; mathieumahamoudissa@hotmail.fr; 3SNSB-Bavarian State Collection of Zoology, Section Ornithology, Münchhausenstr. 21, 81247 Munich, Germany; unsoeld@snsb.de; 4Laboratory and Museum of Evolutionary Ecology, Department of Ecology, Faculty of Humanities and Natural Sciences, University of Prešov, 080 01 Prešov, Slovakia; hromada.martin@gmail.com

**Keywords:** acari, birds, diversity, ectoparasites, quill mites

## Abstract

**Simple Summary:**

This study investigated the picobiine mites parasitising African barbets. The results showed that this bird family is more widely infested by feather mites than previously thought, with three species of the genus *Tanopicobia* found on ten African barbet species. Birds belonging to the family Lybiidae have a unique parasite fauna consisting exclusively of mites from the genus *Tanopicobia*, which is restricted solely to African barbets. Based on the distribution of the genus *Tanopicobia* on the studied barbet hosts, our results also provide indirect cues that the host genus *Trachyphonus*, attributed to such different avian families, e.g., Capitonidae, Ramphastidae, is part of the family Lybiidae, whereas other/related bird families have their own distinct quill mite fauna.

**Abstract:**

In this study, we conducted a parasitological investigation of the quill mite fauna of the subfamily Picobiinae (Acariformes: Prostigmata: Syringophilidae) associated with African barbets (Aves: Piciformes: Lybiidae). We examined twenty-seven host species, representing 57% of the forty-seven known host species in the family Lybiidae, belonging to seven genera (70% of the ten genera in the family). Our research revealed that ten host species were infested by three species of picobiine mites belonging to the genus *Tanopicobia*: (1) *Tanopicobia hallae* Sikora and Skoracki, sp. n., from three species of the genus *Lybius* and two species of the genus *Tricholaema*, (2) *Tanopicobia stactolaema* Sikora and Skoracki, sp. n., from two species of the genus *Stactolaema*, and (3) *Tanopicobia trachyphoni* Skoracki et al., 2020, collected from three host species of the genus *Trachyphonus*. Our findings demonstrate that birds belonging to the family Lybiidae have a specific parasite fauna consisting exclusively of mites of the genus *Tanopicobia*; this mite genus is apparently restricted to African barbets.

## 1. Introduction

Quill mites of the subfamily Picobiinae (Acariformes: Prostigmata: Syringophilidae) are permanent and host-specific ectoparasites of birds. Their whole life cycle takes place inside quills of the contour feathers where they live, feed, and reproduce [1,2,3,4]. The exception is an ambiguous species *Calamincola lobatus* Casto, which inhabits quills of secondaries [5]. Currently, the subfamily comprises c.a. 80 described species grouped in 12 genera and recorded from more than 200 host species belonging to neognathous birds (Aves: Neognathae) [4,6]. The quill mite fauna of picobiines, known from birds of the order Piciformes, are still understudied. Although the first record of picobiine mites from Piciformes was presented more than a hundred years ago when Haller, in 1878, described the first species *Picobia heeri* [7], it is only relatively recently that intensive research on this group of hosts has begun. In several papers, the picobiine fauna have been presented for birds of the families Lybiidae [8], Picidae [1,3,4,9,10,11,12,13,14], Semniornithidae [15], and Ramphastidae [16]. To this time, we have no data on the presence of picobiine mites from birds of the families Megalaimidae, Capitonidae, and Indicatoridae.

In this paper, we present the results of our study on quill mites of the subfamily Picobiinae parasitising African barbets (Lybiidae). This avian family comprises about 41 species grouped in seven genera distributed mainly in sub-Saharan Africa [17,18,19]. The African barbets are medium-sized birds, ranging from 15 to 30 cm in length and are found in a variety of habitats, including forests, woodlands, savannas, and gardens. Some species are more specialised, inhabiting specific habitats such as montane forests or riverine woodlands. Several species of African barbets are considered threatened or endangered due to habitat loss and fragmentation, particularly in West and Central Africa. Overall, the African barbets are a fascinating group of birds that play important ecological roles in African ecosystems [18,19,20]. The Lybiidae family consists of bird species with different social organizations: some species live in single pairs, while other species live in family groups and even small colonies and are considered as group breeding species [21]. It is thus an excellent family to study the evolution of the host–parasite speciation and transmission according to the degree of complexity of the social organization, as well as possible inter-species contamination. Moreover, African barbets are nesting in tree cavities, which also could modify the probability of infestation by ectoparasites, and even more interestingly, they are brood-parasitised by other piciform birds, e.g., Indicatoridae [19].

Because to date, only one species, *Tanopicobia trachyphoni* Skoracki et al., has been recorded from one host species, i.e., the red and yellow barbet, *Trachyphonus erythrocephalus* Cabanis [8], we conducted a parasitological investigation of the picobiine fauna associated with the birds of the family Lybiidae. Our research revealed that ten host species were infested by three species of picobiine mites, including two species described herein as new to science. Our findings demonstrate that birds belonging to the family Lybiidae have a unique parasite fauna consisting exclusively of mites from the genus *Tanopicobia* and that the distribution of this mite genus is restricted solely to African barbets.

## 2. Materials and Methods

The mite material used in this study was collected from dry bird skins housed in the ornithological collection of the Bavarian State Collection of Zoology (Munich, Germany) (by M.S and M.U.), according to the technique described by Skoracki [3] (Figure 1A–D). Additional mite material was collected from the yellow-breasted barbet *Trachyphonus margaritatus* (Cretzschmar) captured during a field expedition in Djibouti (by B.S. and M.M-I) (permit no. 438/DEDD/2020 to M.M-I) (Figure 1E).

We examined the quills of approximately ten contour feathers in the proximity of the cloaca region for each bird. Before mounting, mites were treated in Nesbitt’s solution at room temperature for three days, following the procedure outlined by Walter and Krantz [22] and Skoracki [3] to soften and clear them. Subsequently, the mites were mounted on slides in Hoyer’s medium and examined under a light microscope (ZEISS Axioscope, Oberkochen, Germany) with differential interference contrast (DIC) illumination. To illustrate the findings, we created drawings using a camera lucida drawing attachment. Finally, drawings were made using a camera lucida drawing attachment.

All measurements provided in the description are in micrometers. The paratypes’ dimensional ranges are indicated in parentheses alongside the holotype data. Idiosomal setation adheres to Grandjean’s [23] classification, adapted for Prostigmata by Kethley [24]. Leg chaetotaxy follows the proposal made by Grandjean [25]. Finally, the morphological terminology follows Skoracki [3,4].

Specimen depositories are cited using the following abbreviations: AMU—Adam Mickiewicz University, Department of Animal Morphology, Poznan, Poland; SNSB—ZSM—Bavarian Natural History Collections—Bavarian State Collection of Zoology, Munich, Germany.

## 3. Results

### Systematics

Family Syringophilidae Lavoipierre

Subfamily Picobiinae Johnston and Kethley

Genus *Tanopicobia* Skoracki, Sikora, Jerzak and Hromada, 2020

#### Descriptions

##### *Tanopicobia hallae* Sikora and Skoracki sp. n. (Figure 2)

Female. Total body length of holotype 450 (440–530 in seven female paratypes). *Gnathosoma*. Infracapitulum apunctate. Stylophore 100 (100–110) long; exposed portion of stylophore (stylophoral shield) apunctate, 70 (70–80) long. Each medial branch of peritremes with six chambers, each lateral branch with weakly marked borders between chambers. Movable cheliceral digit edentate on proximal end. *Idiosoma*. Setae *vi*, *ve*, *si*, *se*, *c1*, *c2*, *d1*, *d2*, *e2*, *3b*, *4b*, *3c*, *4c*, and *3a* strongly ornamented. Setae *1a* and *ag1*–*3* smooth. Propodonotal shield divided into three sclerites: two lateral shields bearing bases of setae *si* and *se* narrowly separated from large medial shield bearing bases of setae *vi*, *ve*, and *c1*; all propodonotal sclerites punctate. Length ratio of setae *vi*:*ve*:*si* 1:1.6:2.2–2.4. Hysteronotal shield reduced to two well developed and punctate sclerites surrounding bases of setae *d1*. Hysteronotal setae *d1*, *d2*, and *e2* subequal in length. Pygidial shield present, well sclerotised and punctate, 95 (90–95) long. Setae *f2* 3.5–4 times longer than *f1*. Genital plate present, punctate. Pseudanal setae as microsetae. Setae *ag1* 3.3–4 times longer than *ag2*. Coxal fields I–II apunctate, III and IV punctate. Setae *3c* 1.6 times longer than *3b*, *4c* about twice as long as *4b*. *Legs*. Setae *dFI*, *dGI*, *dTI*, *l’GI*–*IV*, *l’TI*–*IV*, and *l’RIII*–*IV* strongly knobbed, other leg setae slightly ornamented or smooth.

**Figure 2 animals-13-02007-f002:**
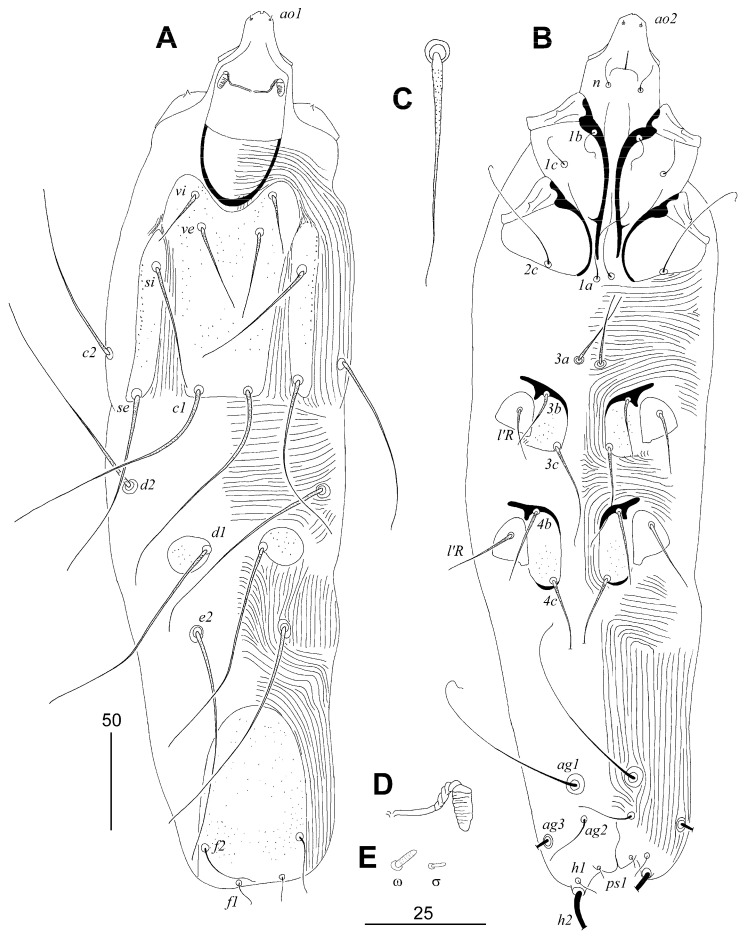
*Tanopicobia hallae* Sikora and Skoracki sp. n., female. (**A**)—dorsal view; (**B**)—ventral view; (**C**)—propodonotal seta *ve*; (**D**)—peritreme; (**E**)—solenidia of leg I. Scale bars: (**A**,**B**) = 50 µm, (**C**–**E**) = 25 µm.

Lengths of setae: *vi* 30 (30–35), *ve* 45 (45), *si* 70 (50–70), *se* 95, *c1* 125 (120–130), *c2* 95 (95–100), *d1* 115 (105–115), *d2* 115 (110–115), *e2* 125 (100–125), *f1* (10), *f2* 30 (30–40), *h1* (10), *h2* (225), *ag1* 105 (100–105), *ag2* 25 (25–30), *ps1* (5), *3a* 40 (40), *3b* 30 (30–40), *3c* 50 (50–65), *4b* 40 (40), *4c* (75–80), *l’RIII* 30 (30–35), *l’RIV* 35 (30–35), *tc’III*–*IV* (25), *tc”III*–*IV* (45).

Male. Not found.

Type Material

Female holotype and seven female paratypes from quill of contour feathers of the banded barbet *Lybius undatus* (Rüppell) (host reg. no. SNSB-ZSM 66.200, female); ETHIOPIA: Benishangul-Gumuz Region, Lekamti, 15–17 November 1965, coll. K.E. Linsenmair.

Type Material Deposition

Holotype and paratypes are deposited in the AMU (reg. no. MS 22-1022-053), except three female paratypes in the SNSB-ZSM (reg. no. SNSB-ZSM A20112197).

Additional material

Ex quill of contour feather of the black-collared barbet *Lybius torquatus* (Dumont) (host reg. no. SNSB-ZSM 1287; male); TANZANIA: Morogoro Region, Morogoro District, 27 November 1952, coll. Th. Andersen—four females deposited in AMU (reg. no. AMU MS 22-1022-052a) and two females deposited in SNSB-ZSM (reg. no. SNSB-ZSM A20112198).

Ex quill of contour feather of the brown-breasted barbet *Lybius melanopterus* (Peters) (host reg. no. SNSB-ZSM 60.1784; female); TANZANIA: Arusha Region, Arusha District, Usa-River, 22 February 1960, coll. unknown—five females deposited in AMU (reg no. AMU MS 22-1022-054) and two females in SNSB-ZSM (reg. no. SNSB-ZSM A20112198).

Ex quill of contour feather of the spot-flanked barbet *Tricholaema lacrymosa* Cabanis (host reg. no. SNSB-ZSM 64.894; female); TANZANIA: Lindi Region, Kilwa District, 18 July 1952, coll. Th. Andersen—4 females deposited in AMU (reg. no. AMU MS 22-1022-119) and 4 females in SNSB-ZSM (reg. no. SNSB-ZSM A20112205).

Ex quill of contour feather of the red-fronted barbet *Tricholaema diademata* (Heuglin) (host reg. no. SNSB-ZSM 60.1774; male); TANZANIA: Arusha Region, Arusha District, near Arusha, 14 March 1960, coll. von Nagy—three females and one male deposited in AMU (reg. no. AMU MS 22-1022-122) and three females in the SNSB-ZSM (reg. no. SNSB-ZSM A20112204).

Differential Diagnosis

This new species differs from *T. trachyphoni* as follows: in females of *T. hallae*, the lengths of setae *ve* and *si* are 45 and 50–70, respectively; and the hysteronotal shields are well-developed and punctate. In females of *T. trachyphoni*, the lengths of setae *ve* and *si* are 70–80 and 95–110, respectively; and the hysteronotal shields are absent.

Etymology

This species is named in honour of the British ornithologist Dr. Beryl Patricia Hall (1917–2010), an expert in the distribution and speciation of African birds.

##### *Tanopicobia stactolaema* Sikora and Skoracki, sp. n. (Figure 3 and Figure 4)

Female (Figure 3). Total body length 430 in holotype (460–525 in six paratypes). *Gnathosoma*. Stylophore 115 (115) long; exposed portion of stylophore (stylophoral shield) apunctate, 80 (80) long. Each medial branch of peritremes with six or seven chambers, each lateral branch with weakly marked borders between chambers. Movable cheliceral digit edentate on proximal end. *Idiosoma*. Setae *vi*, *ve*, *si*, *se*, *c1*, *c2*, *d1*, *d2*, *e2*, *3b*, *4b*, *3c*, *4c*, and *3a* strongly ornamented. Setae *1a* and *ag1*–*3* smooth. Propodonotal shield divided into three sclerites: two lateral shields bearing bases of setae *si* and *se* narrowly separated from large medial shield bearing bases of setae *vi*, *ve*, and *c1*; all propodonotal sclerites apunctate. Length ratio of setae *vi*:*ve*:*si* 1:1.6–1.9:2–2.6. Two hysteronotal shields well developed and apunctate, posterior margin of each shield reaching bases of setae *e2*. Hysteronotal setae *d1*, *d2*, and *e2* subequal in length. Pygidial shield present, well sclerotised and apunctate. Setae *f2* slightly (1.3 times) longer than *f1*. Setae *ag1* 2.8–3.2 times longer than *ag2*. Coxal fields I–IV apunctate. Setae *3c* and *4c* 1.3–1.6 times longer than *3b* and *4b*. *Legs*. Setae *dFI*, *dGI*, *dTI*, *l’GI*–*IV*, *l’TI*–*IV*, and *l’RIII*–*IV* strongly knobbed, other leg setae slightly ornamented or smooth.

**Figure 3 animals-13-02007-f003:**
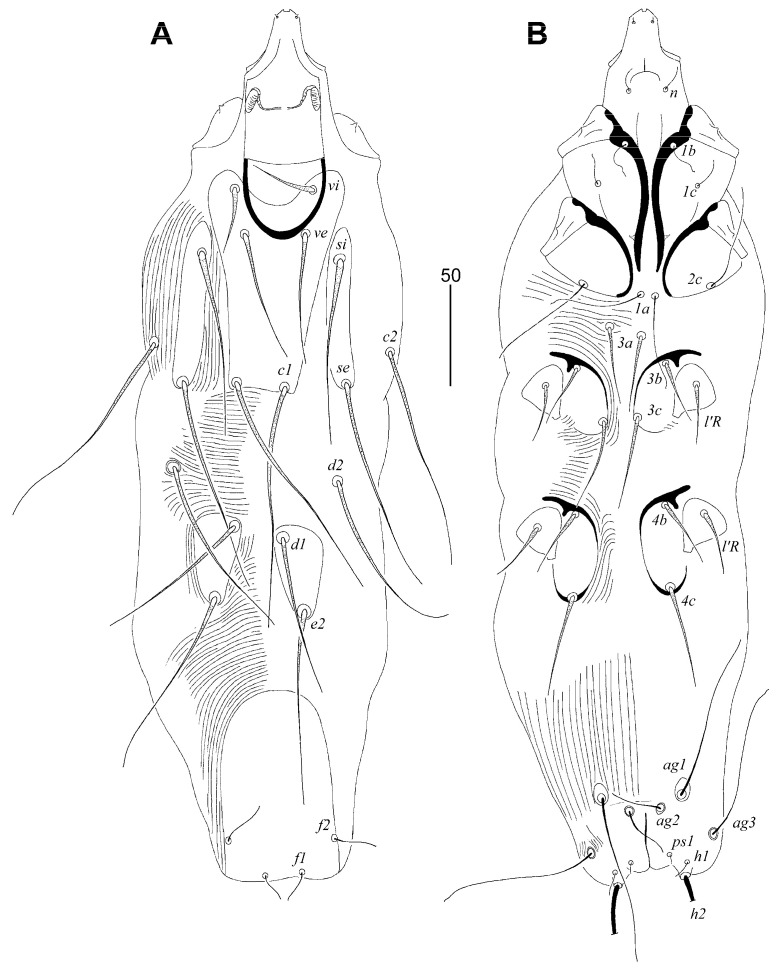
*Tanopicobia stactolaema* Sikora and Skoracki sp. n., female. (**A**)—dorsal view; (**B**)—ventral view.

**Figure 4 animals-13-02007-f004:**
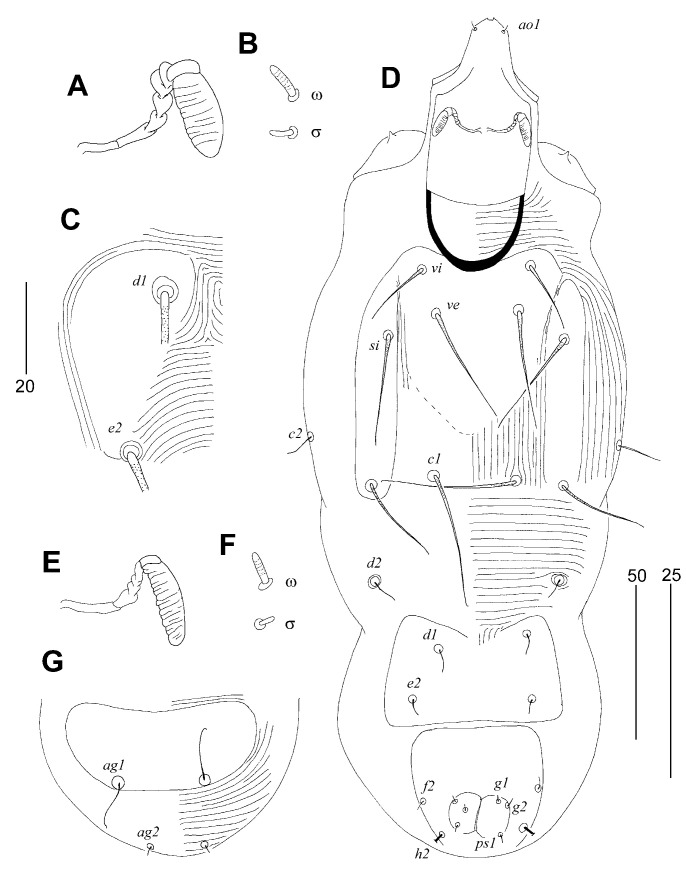
*Tanopicobia stactolaema* Sikora and Skoracki sp. n. Female (**A**–**C**). (**A**)—peritreme; (**B**)—solenidia of leg I; (**C**)—left hysteronotal shield. Male (**D**–**G**). (**D**)—dorsal view; (**E**)—peritreme; (**F**)—solenidia of leg I; (**G**)—opisthosoma in ventral view. Scale bars: (**A**,**B**,**E**,**F**) = 25 µm; (**C**) = 20 µm; (**D**, **G**) = 50 µm.

Lengths of setae: *vi* 35 (35–45), *ve* 65 (60–70), *si* 90 (80–90), *se* 110 (110–120), *c1* 115 (115), *c2* 110 (95–115), *d1* (70–80), *d2* 95 (80–95), *e2* 95 (90–95), *f1* 20 (15–20), *f2* 25 (25), *h1* 10 (10–15), *h2* (205–235), *ag1* (85–95), *ag2* 30 (20–30), *ag3* 70 (65–70), *ps1* 5 (5), *3a* 40 (40), *3b* 35 (35), *3c* 45 (45–55), *4b* 35 (35), *4c* 45 (45–55), *l’RIII* 35 (30–35), *l’RIV* 35 (30–35), *tc’III*–*IV* 30 (30), *tc”III*–*IV* 45 (45).

Male (Figure 4). Total body length 275 in one paratype. *Gnathosoma*. Infracapitulum apunctate. Stylophore 80 long; exposed portion of stylophore apunctate, 65 long. Each medial branch of peritremes with six chambers, each lateral branch with weakly marked borders between chambers. *Idiosoma*. Propodonotal setae *vi*, *ve*, *si*, *se*, and *c1* strongly ornamented. Setae *1a* and *ag1*–*2* smooth. Propodonotal shield divided into three apunctate sclerites: two lateral shields bearing bases of setae *si* and *se*, both shields narrowly separated from large medial shield bearing bases of setae *vi* and *ve*. Setae *ve* and *si* twice as long as *vi*. Hysteronotal shield apunctate, bearing bases of setae *d1* and *e2*. Pygidial shield apunctate. Hysteronotal setae *d1*, *d2*, *e2*, and *f2* short and smooth. Aggenital plate entire and apunctate, bearing bases of setae *ag1* on posterior margin of this plate. Setae *ag1* distinctly longer than *ag2*. Coxal fields I–IV apunctate. Setae *3b*, *4b*, *3c*, *4c*, and *3a* strongly ornamented.

Lengths of setae: *vi* 20, *ve* 40, *si* 40, *se* 30, *c1* 45, *c2* 15, *d1* 8, *d2* 8, *e2* 5, *f2* 4, *ag1* 15, *ag2* 4.

Type Material

Female holotype, six female paratypes from quill of contour feather of the white-eared barbet *Stactolaema leucotis* (Sundevall) (host reg. no. SNSB-ZSM 59.257; female); TANZANIA: Arusha Region, Arusha District, Arusha, 10 February 1958, coll. von Nagy. One female paratype from same host species (host reg. no. SNSB-ZSM uncatalogued; female); TANZANIA: Kilimanjaro Region, Same District, 9 February 1954, coll. Th. Andersen. Two female and one male paratypes from same host species (host reg. no. SNSB-ZSM 60.31; female); TANZANIA: Arusha Region, Arusha District, Usa-River, 6 December 1959, coll. unknown.

Type Material Deposition

Holotype and paratypes are deposited in the AMU (reg. no. AMU MS 22-1022-114/115/116), except two female paratypes in SNSB-ZSM (reg. no. SNSB-ZSM A20112200).

Additional Material

Ex quill of contour feather of the green barbet *Stactolaema olivacea* (Shelley) (host reg. no. SNSB-ZSM uncatalogued; male); TANZANIA: Morogoro Region, Morogoro District, 5 March 1955, coll. Th. Andersen—three females in AMU (reg. no. AMU MS 22-1022-110) and four females in SNSB-ZSM (reg. no. SNSB-ZSM A20112201).

Differential Diagnosis

*Tanopicobia stactolaema* sp. n. is morphologically similar to the above-described species, *T. hallae* sp. n., by the presence of the well-visible hysteronotal shields and is distinguishable by the following features: in females of *T. stactolaema*, the propodonotal, hysteronotal, and pygidial shields are apunctate; the posterior margins of the hysteronotal shields reach bases of setae *e2*; the lengths of propodonotal setae *ve* and *si* are 60–70 and 80–90, respectively; the lengths of hysteronotal setae *d1* and *d2* are 70–80 and 80–95, respectively; setae *f2* are slightly (1.3 times) longer than *f1*; coxal fields III and IV are apunctate, and setae *4c* are 45–55 long. In females of *T. hallae*, the propodonotal, hysteronotal, and pygidial shields are punctate; the posterior margins of the hysteronotal shields not reach bases of setae *e2*; the lengths of propodonotal setae *ve* and *si* are 45 and 50–70, respectively; the lengths of hysteronotal setae *d1* and *d2* are 105–115 and 110–115, respectively; setae *f2* are 3.5–4 times longer than *f1*; coxal fields III and IV are punctate, and setae *4c* are 75–80 long.

Etymology

The specific name “*stactolaema*” is taken from the generic name of the host.

##### *Tanopicobia trachyphoni* Skoracki, Sikora, Jerzak and Hromada, 2020 (Figure 5)

This species was recently described from the red-and-yellow barbet *Trachyphonus erythrocephalus* Cabanis in Tanzania [8], and there have been no other records since the first description. Herein, we report two new hosts species for this quill mite: the yellow-breasted barbet, *Trachyphonus margaritatus* (Cretzschmar), from Djibouti and Eritrea and the d’Arnaud’s barbet *Trachyphonus darnaudii* (Prévost and des Murs), from Tanzania.

Because this species was described based only on the holotype and two female paratypes, herein, we give additional data for its description based on the material collected from the type of host species, *T. erythrocephalus*, from Tanzania.

**Figure 5 animals-13-02007-f005:**
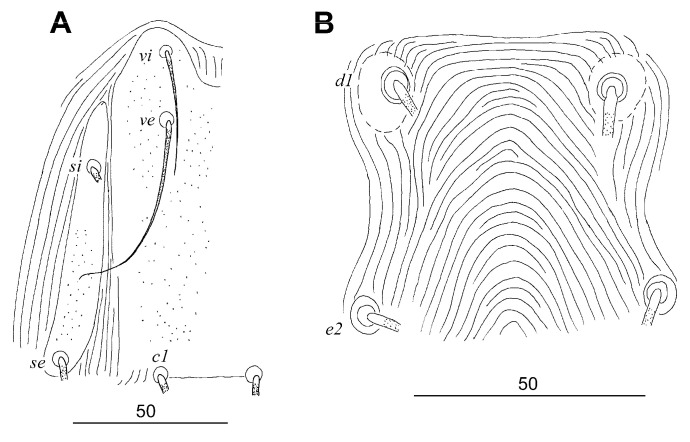
*Tanopicobia trachyphoni* Skoracki et al., 2020, female. (**A**) Propodonotal shield; (**B**) hysteronotum with hysteronotal shields around bases of setae *d1*.

Complementary Description

Female (14 specimens from *T. erythrocephalus* from Tanzania; measurements of type material [8] are in square brackets). Propodonotal shield entire or lateral sclerites bearing bases of setae *si* and se narrowly separated from large medial shield bearing bases of setae *vi*, *ve*, and *c1*. Length of stylophore and stylophoral shield 110–115 and 80–85, respectively. Lengths of setae: *vi* 35–40 [40], *ve* 75–80 [70–80], *si* 95–110 [95], *se* 95–115 [100–110], *c1* 140–155 [140–145], *c2* 110–130 [115], *d1* 110–130 [110–120], *d2* 110–130 [115], *e2* 145 [115–120], *f1* 5–7 [7], *f2* 15–20 [25–35], *h1* 5–7 [10], *h2* 220–265 [260], *ag1* 90 [120–125], *ag2* 25–35 [35], *ag3* 125 [130–140], *ps1* 5 [5], *3a* 30, *3b* 35–40 [40], *3c* 50–55 [50–55], *4b* 35–40 [40], *4c* 50 [50–55], *l’RIII* 35 [35], *l’RIV* 35, *tc’III*–*IV* 25–30 [30], *tc”III*–*IV* 60–70 [70].

New Material Examined

Ex the red-and-yellow barbet *Trachyphonus erythrocephalus* Cabanis; TANZANIA: Arusha Region, Arusha District, S. Arusha, 25 March 1960, coll. von Nagy—four females and one male deposited in the AMU (reg. no. AMU MS 22-1022-033) and five females in the SNSB-ZSM (SNSB-ZSM A20112200). From same host species; TANZANIA: Kilimanjaro Region., Same Distr., Lembani, 25 January 1954, coll. Th. Andersen—two females and two males deposited in the AMU (reg. no. AMU MS 22-1022-034) and three females in the SNSB-ZSM (SNSB-ZSM A20112201).

Ex the yellow-breasted barbet, *Trachyphonus margaritatus* (Cretzschmar) (new host species); ERITREA: Northern Red Sea Region, Massawa, 18–26 December 1965, coll. K. E. Linsenmair—six females deposited in the AMU (reg. no. AMU MS 22-1022-(035-036)) and three females in the SNSB-ZSM (reg. no. SNSB-ZSM A20112195). From same host species; DJIBOUTI: Assamo, 2–6 February 2020, coll. B. Sikora and M. Mahamoud-Issa—thirteen females and two males deposited in the AMU (reg. no. AMU MS 22-1114-(001-008))

Ex the d’Arnaud’s barbet *Trachyphonus darnaudii* (Prévost and des Murs) (new host species); TANZANIA: Manyara–Arusha Region, 13–30 November 1959, coll. von Nagy—ten females and two males (reg. no. AMU MS 22-1022-(037-039, 045)). Ex same host species; TANZANIA: Manyara Region, Babati District, Magugu, 25 July–6 August 1960, coll. von Nagy—four females and one male deposited in the AMU (reg. no. AMU MS 22-1022-(040-042, 046) and three females in the SNSB-ZSM (reg. no. SNSB-ZSM A20112196). Ex same host species; TANZANIA: Dodoma Region, Bahi District 15 June 1953, coll. Th. Andersen—one female deposited in the AMU (reg. no. AMU MS 22-1022-043). Ex same host species; TANZANIA: Dodoma Region, Kondoa District, Busi, 22 August 1960, coll. von Nagy—three females deposited in the AMU (reg. no. AMU MS 22-1022-047).

## 4. Discussion

To date, only one species, *Tanopicobia trachyphoni* Skoracki et al., has been recorded from one host species, the red and yellow barbet, *Trachyphonus erythrocephalus* Cabanis [8]. Herein, we conducted a parasitological investigation of the picobiine quill mite fauna associated with birds of the family Lybiidae. Above, we have demonstrated that this bird family is much more widely infested by quill mites than previously thought [8]. Our research has identified the presence of three species of picobiine mites of the genus *Tanopicobia* on ten African barbets belonging to the four genera, i.e., *Trachyphonus* (three species), *Tricholaema* (two species), *Lybius* (three species), and *Stactolaema* (two species). Our findings presented in this study, demonstrate that birds belonging to the family Lybiidae have a unique parasite fauna consisting exclusively of mites from the genus *Tanopicobia* (lack members of the other picobiine genera) and that the distribution of this mite genus is restricted solely to the African barbets. Unfortunately, the small sample size of individuals examined from the genera *Gymnobucco*, *Pogoniulus*, and *Buccanodon* has not allowed us to confirm the presence of mites on these birds, but we are rather confident that future studies will demonstrate the occurrence of mites on these bird genera. Additionally, given that all examined bird genera of the African barbets were infested by members of the genus *Tanopicobia*, it is expected that species (supposedly new to science) of this genus will also be present on these birds.

*Systematic position of Trachyphonus birds* vs. *quill mites.* The genus *Trachyphonus* Ranzani, comprises five species distributed exclusively in sub-Saharan Africa [17,20]. Although *Trachyphonus* is currently classified in the family Lybiidae, the relationship of this genus seems to be one of the most intricate problems in the barbet phylogeny. Swierczewski and Raikow [26], using variations in the hind limb muscle, and Bellman [27], analysing fossil records, proposed that *Trachyphonus* is the sister group to the rest of the species in the family Capitonidae. Later, Prum [28] used a cladistic analysis of morphological characters and placed the representatives of *Trachyphonus* in the newly erected subfamily Trachymphoninae in the family Ramphastidae. In 2000, Barker and Lanyon used mitochondrial DNA sequence data [29] and placed it as the sister taxon to the Neotropical radiation. In contrast, in 2004, Moyle, based on the combined gene analyses, placed *Trachyphonus* as the basal taxon of the African radiation and indicated that African barbets (Lybiidae) are monophyletic [30]. Moreover, *Trachyphonus* is considered to be an old and early diverging lineage [26,28,30,31,32], which may not even be closely related to other African barbets [30]. Our parasitological investigation of the picobiine mites associated with African barbets provides indirect but rather supportive evidence that birds of the genus *Trachyphonus* are indeed part of the family Lybiidae, as they host the same quill mites of the genus *Tanopicobia* as other members of this bird family. In contrast, the other families of the order Piciformes have their own distinct quill mite fauna, e.g., mites of the genera *Picobia* and *Neopicobia* infest birds in the family Picidae, *Pseudopicobia* infests birds in the family Bucconidae, and *Rafapicobia* infests birds in the families Semniornithidae and Ramphastidae [1,3,4,8,9,10,11,12,13,14,15,16].

## 5. Conclusions

This study investigated the quill mites on birds of the family Lybiidae (African barbets). The results showed that this bird family is more widely infested by feather mites than previously thought, with three *Tanopicobia* species found on ten African barbet species. Birds belonging to the family Lybiidae have a unique parasite fauna consisting exclusively of mites from the genus *Tanopicobia*, which is restricted solely to African barbets. The study also provides indirect evidence that birds of the genus *Trachyphonus* are indeed part of the family Lybiidae based on their hosting of *Tanopicobia* quill mites, while other bird families have their own distinct quill mite fauna.

## Figures and Tables

**Figure 1 animals-13-02007-f001:**
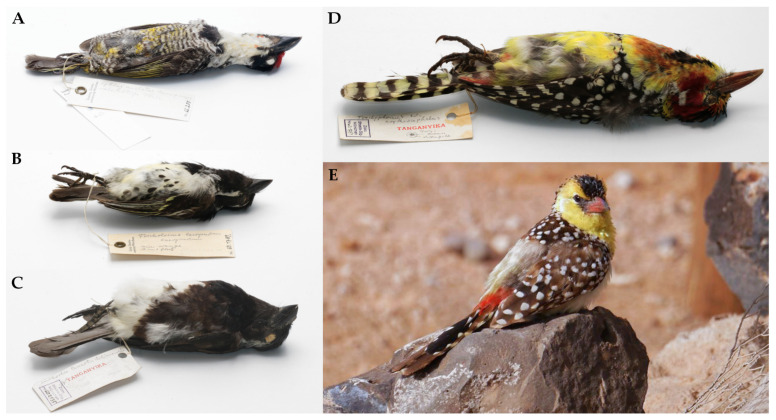
Representatives of the host genera of the family Lybiidae infested by picobiine mites. (**A**)—the banded barbet *Lybius undatus* (Rüppell); (**B**)—the spot-flanked barbet *Tricholaema lacrymosa* Cabanis; (**C**)—the white-eared barbet *Stactolaema leucotis* (Sundevall); (**D**)—the red-and-yellow barbet *Trachyphonus erythrocephalus* Cabanis; (**E**)—the yellow-breasted barbet, *Trachyphonus margaritatus* (Cretzschmar). Photos: (**A**–**D**) M.U.; (**E**) M.M-I.

## Data Availability

Data are available upon request from the corresponding author.

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
