# Peer review of "Species Composition of Parasitic Mites of the Subfamily Picobiinae (Acariformes: Syringophilidae) Associated with African Barbets (Piciformes: Lybiidae)â€"

_animals, 2023, doi:10.3390/ani13122007_

Round 1
Reviewer 1 Report
The reviewed manuscript is worth and useful work enlarging our knowledge on the diversity, host associations and distribution of quill mites (Syringophilidae) and obviously deserves to be published. The paper is well written and needs only minor revisions, mainly re-editing of some phrases and used terms. All comments, questions and proposed corrections are in the attached text file. Several important comments are provided below.
Lines 30-32 (Abstract) and 75-77 (Introduction).
The final phrase in these paragraphs “Our findings demonstrate that birds belonging to the family Lybiidae have a unique parasite fauna consisting exclusively of mites from the genus Tanopicobia and that the distribution of this mite genus is restricted solely to African bar bets” should be better re-edit as follow:
Our findings demonstrate that birds belonging to the family Lybiidae have a specific parasite fauna consisting exclusively of mites of the genus Tanopicobia and this mite genus is apparently restricted to African barbets.
Lines 109 and 110.
It is necessary to add a heading between these lines with the words “Genus Tanopicobia Skoracki, Sikora, Jerzak and Hromada, 2020”
Lines 118, 191. “Movable cheliceral digit edentate on proximal end”
Apparently in both cases, the authors mean the “distal end”, i.e. the tip or apex of the movable digit, which is used for piercing or boring the quill wall.
Paragraphs “Hosts and Distribution” (lines 139-144)
These paragraphs looks excessive in the manuscript, because they briefly repeat the information given in the corresponding paragraphs “Type material”, “Additional material”.
Line 177. Diff. diagnosis,. “… in females of T. lybiusi, …”
Wrong species name, apparently the authors mean T. hallae.
Line 284, 285. "Supplementary Description. Female (ten specimens from T. erythrocephalus from Tanzania …"
“Supplementary Description” seems to be not correct heading here. The authors give new ranges for measured characters, based on a new material, i.e. some additions to the description, not a completely renewed description. It is more correct to write “Supplementary data for description”.
The authors write that they measured ten newly collected females of T. erythrocephalus. However, the paragraphs “New Material Examined” gives 14 females from this host in Tanzania (see lines 296-302). If the authors measured only 10 of 14 females, it should be exactly said in the line 285.
Line 288. “… Length of stylophore and stylophoral shield 110–115 and 80–85, respectively.”
In the lines 285, 286, it is said that the measurements for mites from the type series are given in the rectangular brackets after the measurements for new specimens. These new measurements for the stylophore and stylophoral shield are not supplemented by the measurements of the type specimens, in contrast to subsequent measurements for setae (lines 289-293).

The English in the MS is good and clear, just a few phrases need re-edition (see the attached file).
The author use the word "herein" too often. It seems to be excessive.
Author Response
Dear Reviewer,
thank you very much for your remarks and corrections to our manuscript. We have taken into consideration all suggestions except:
Point 1. Movable cheliceral digit edentate on proximal end” Apparently in both cases, the authors mean the “distal end”, i.e. the tip or apex of the movable digit, which is used for piercing or boring the quill wall.
Response 1. We didn't mean tip of the chelicerae but the proximal end (please see Skoracki et al. 2016: page 8 Fig 3F). The text is unchanged.
Point 2. In the lines 285, 286, it is said that the measurements for mites from the type series are given in the rectangular brackets after the measurements for new specimens. These new measurements for the stylophore and stylophoral shield are not supplemented by the measurements of the type specimens, in contrast to subsequent measurements for setae (lines 289-293).
Response 2. The measurements for the stylophore and stylophoral shield are not given in the original description.
All the best,
Maciej Skoracki
Reviewer 2 Report
This is an important contribution to the knowledge of a poorly-known group of parasitic mites. As all other species descriptions, it is basic science which will make other studies possible in the future.
The authors are internationally recognized experts in this field.
Author Response
Dear Reviewer,
thank you for your opinion about our manuscript.
All the best,
Maciej Skoracki
Reviewer 3 Report
The research concerns the analysis of the occurrence of mites of the subfamily Picobiinae (Acariformes: Syringophilidae) in African barbets and brings a great deal of valuable new data. They should undoubtedly be published, with minor corrections.
Comments
Materials and Methods - The number of birds (hosts) studied was not stated. In the abstract, it is stated what number of host species was included and it was a very representative sample. But in the description of the study material, details (list of study species, number of specimens) should have been given. Perhaps it would also be useful to state in the Results how many of the studied birds were found to have mites (basic parasitological data)? Although the study is of a qualitative nature (analysis of species diversity, new parasite-host relationships), the sample size, even with a small number of specimens tested, illustrates the level of infestation to some extent and is important for parasitological analyses (also in the context of the Discussion, e.g. lines 332-335).
Author Response
Dear Reviewer,
thank you for suggesting that we add the number of examined host specimens. These data will be presented in the future paper about host-parasite relationships of the system composed of quill mites and all groups of barbets (Lybiidae, Megalaimidae, Capitonidae).
All the best,
Maciej Skoracki